# Ampere-level reduction of pure nitrate by electron-deficient Ru with K$^+$ ions repelling effect

Shi-Nan Zhang [1,2], Peng Gao [1,2], Qian-Yu Liu[1], Zhao Zhang[1], Bing-Liang Leng [1], Jie-Sheng Chen [1] & Xin-Hao Li [1] ✉

Electrochemical nitrate reduction reaction offers a sustainable and efficient pathway for ammonia synthesis. Maintaining satisfactory Faradaic efficiency for long-term nitrate reduction under ampere-level current density remains challenging due to the inevitable hydrogen evolution, particularly in pure nitrate solutions. Herein, we present the application of electron deficiency of Ru metals to boost the repelling effect of counter K$^+$ ions via the electric-field-dependent synergy of interfacial water and cations, and thus largely promote nitrate reduction reaction with a high yield and well-maintained Faradaic efficiency under ampere-level current density. The pronounced electron deficiency of Ru metals boosts the repelling effect on hydrated K$^+$ ions, as indicated by the distance of K$^+$ ions to catalyst surface, which can loosen the water layer to depress hydrogen evolution and accelerate nitrate conversion. Consequently, the optimized electrode loaded with electron-deficient Ru atomic layers can directly produce 0.26 M ammonia solution in pure nitrate solution in 6 h, providing a high yield (74.8 mg mg$_{cat}^{-1}$ h$^{-1}$) and well-maintained the Faradaic efficiency for over 120 h under ampere-level reduction.

Ammonia (NH$_3$) not only acts as a pivotal component in nitrogenous fertilizers production but also holds great potential as a promising clean energy carrier and storage media of hydrogen owing to its high energy density and hydrogen content[1,2]. As compared with the Haber−Bosch process, electrochemical nitrate reduction reaction (NO$_3$RR) powered by renewable electricity offers a win-win opportunity for ammonia synthesis and energy storage under ambient conditions[3–5]. To meet the demands of real applications via NO$_3$RR process, the mainstream electrocatalysts have been recently designed in electrolyzers using various electrolyte solutions to get a current density higher than 1.0 A cm$^{-2}$ and also high Faradic efficiency (FE) at the initial stage[6,7]. However, it is quite challenging to maintain satisfactory FE values with the evolution of NO$_3$RR towards ammonia in long-term uses accompanied by more pronounced hydrogen evolution process[8,9], especially in additive-free nitrate solutions[10]. Moreover, industrial scaling of NO$_3$RR requires efficient conversion of concentrated nitrates for ammonia synthesis. Achieving durable ammonia synthesis with satisfactory yields and FE values at high concentrations of pure nitrates and thus directly obtaining concentrated ammonia solutions to reduce additional separation costs remains a substantial challenge[7,11].

Developing more powerful electrocatalysts/electrodes for long-term uses[12] in NO$_3$RR mainly relies on the mechanism study on interactions of interfacial water and ions[13,14] in the electric double layer to depress the competitive reactions (e.g. hydrogen evolution reaction (HER)). Recently, the cation effect in electrocatalysis has been paid much attention, demonstrating an effective way to tune the microenvironment via interactions between interfacial water and cations[15,16]. Pioneering studies have revealed how different cations (such as Li$^+$, Na$^+$, and K$^+$) affect the structure of interfacial water to boost the electrocatalytic performance for various electrocatalytic reactions[17–19], while effective method to control the distribution of

[1]School of Chemistry and Chemical Engineering, Frontiers Science Center for Transformative Molecules, Shanghai Jiao Tong University, Shanghai 200240, P. R. China. [2]These authors contributed equally: Shi-Nan Zhang, Peng Gao. ✉e-mail: xinhaoli@sjtu.edu.cn

specific cations and the interaction with interfacial water has been rarely touched.

Herein, we present the proof-of-concept application of electron deficiency of Ru metals to boost the cation repelling effect (exemplified with K+ in this work) via the electric-field-dependent synergy of interfacial water and cations, and thus largely promote NO₃RR with a high and durable FE under ampere-level current density. Experimental results in the literature[7,9] and from our group[10] have shown that Ru metal-based electrocatalyst could provide very high current output for NO₃RR and was thus selected as the model catalyst with electron deficiency effect for possible application in the ampere-level nitrate reduction. A special electronic interface-induced strategy was applied to control the synthesis of two-dimensional (2D) Ru on a rationally designed nitrogen-doped carbon support (2D-Ru/NC) to construct a rectifying interface and achieve high electron deficiency. In situ Raman spectra and Ab initio molecular dynamics (AIMD) results reveal that the electron-deficient surface of 2D Ru metals exhibited a unique repelling effect on the hydrated K+ ions in the electric double layer, as indicated by the distance of K+ ions to catalyst surface, which could loosen the water layer to depress HER and accelerate the penetration

of NO₃− ions to give a fast conversion of nitrate. Consequently, the optimized 2D-Ru/NC electrode with the most pronounced electron deficiency of Ru (loss of 0.04 e− per Ru atom) could produce 0.26 M ammonia solution in pure nitrate solution in 6 h, achieving a high yield (74.8 mg mg$_{cat}^{-1}$ h$^{-1}$) and most importantly maintaining the FE values for over 120 h under ampere-level reduction.

## Results

### Preparation and characterization of 2D-Ru/NC catalyst

The synthesis of 2D-Ru/NC heterojunction followed a reported method[20], which started with the synthesis of the atomically dispersed Ru (a-Ru/NC) as the pre-catalyst through the pre-adsorption of Ru ions on NC support with Cl− as the counter anions. Then the in situ electrochemical reduction of the a-Ru/NC-based electrode was conducted under linear sweep voltammogram (LSV) scanning in an Ar-saturated 0.5 M H₂SO₄ electrolyte (Supplementary Fig. 1). The as-formed Ru metals along the whole NC support were revealed through the aberration-corrected environmental scanning transmission electron microscope dark field[21] (ESTEM-DF) (Fig. 1a and Supplementary Fig. 2) and false-color images (Fig. 1b and Supplementary Fig. 3). The uniform

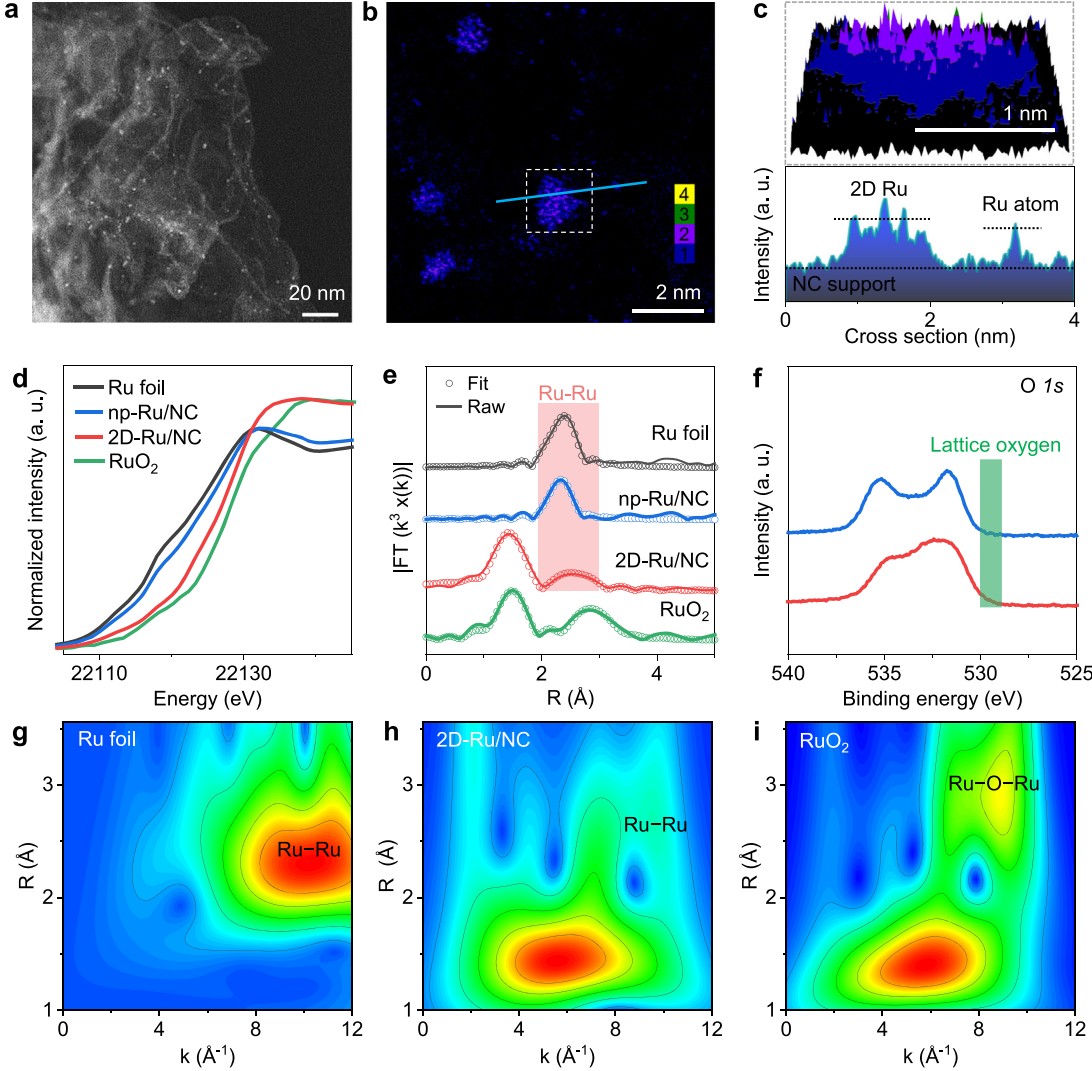

**Fig. 1 | Structural characterization of the 2D-Ru/NC catalyst. a** ESTEM-DF image of 2D-Ru/NC. **b** False-color ESTEM-DF images of as-obtained 2D-Ru/NC catalyst. **c** 3D false-color ESTEM-DF image (white square of (**b**)) and the relative 2D-Ru Z-contrast intensity compared to atomic Ru (blue line of (**b**)) of the 2D-Ru/NC sample from the selected area of (**b**). **d, e** Ru K-edge XANES spectra (**d**) and k³-weighted Ru K-edge EXAFS spectra and corresponding curve-fitting results (**e**) of Ru foil, np-Ru/NC, 2D-Ru/NC, and RuO₂ samples. **f** O 1s XPS spectra of np-Ru/NC and 2D-Ru/NC samples. **g–i** Wavelet transforms for the k³-weighted EXAFS spectra of Ru foil (**g**), 2D-Ru/NC (**h**), and RuO₂ (**i**) samples. a. u.: arbitrary units. Source data for Fig. 1d–f are provided as a Source Data file.

$Z$-contrast intensity in the ESTEM secondary electron[22,23] (ESTEM-SE) images (Supplementary Fig. 4), showing little difference from that of the NC support, confirms the thin nanolayer structure of the Ru metal in 2D-Ru/NC catalyst. Moreover, 3D false-color ESTEM-DF analysis results (Fig. 1c top) clearly present the 2D structures of the resulting Ru metals with the DF intensity of 2D Ru comparable to that of atomic Ru (Fig. 1c bottom and Supplementary Fig. 5), indicating the monolayer structure of Ru metals.

The chemical structure of the Ru nanolayer in 2D-Ru/NC catalyst was further investigated using X-ray absorption fine structure (XAFS) analysis technique[24–26]. The X-ray absorption near-edge structure spectra (XANES) of Ru K-edge (Fig. 1d) of 2D-Ru/NC exhibit an absorption threshold position lower than that of $RuO_2$ but higher than those of Ru foil and Ru nanoparticle-based sample (np-Ru/NC, Supplementary Figs. 6 and 7), indicating the formation of reduced Ru components in 2D-Ru/NC sample with obvious electron deficiency. Indeed, X-ray photoelectron spectroscopy (XPS) results (Supplementary Fig. 8a) further confirm the formation of Ru metals during the electrochemical reduction process. The negligible Cl $2p$ XPS peak (Supplementary Fig. 8b) also reveals the successful removal of $Cl^-$ to release the Ru surface without apparent changes in the composition of the NC support (Supplementary Fig. 9). The O $1s$ XPS spectra (Fig. 1f) further reveal the presence of oxygen species mainly in the form of weakly adsorbed water, carbon dioxide, etc.[27], rather than detectable amounts of the lattice oxygens with typical signal at 529–530 eV[28] (green bar in Fig. 1f).

The $k^3$-weighted Ru K-edge R-space extended XAFS (EXAFS) spectra (Fig. 1e) and corresponding wavelet transform of the EXAFS spectra (Fig. 1g–i and Supplementary Table 2) directly reveal the formation of Ru–Ru bonds in 2D-Ru/NC sample with the co-existence of even more Ru–X bonds. It should be noted that the X components can be replaced with C, N, or O-based adsorbates[29,30] (Supplementary Figs. 11 and 12 and Supplementary Table 2), including graphite carbon, $O_2$, $H_2O$, hydroxyl, pyridine nitrogen, and pyrrole nitrogen, to get well-fitted results (red circles of Fig. 1e by using pyridine nitrogen for fitting), whilst Cl is not suitable counter ion for curve fitting (Supplementary Fig. 13 and Supplementary Table 2). Surprisingly, the estimated coordination number of Ru–Ru bond in the 2D-Ru/NC is as low as 2.4 as compared with np-Ru/NC (10.3) and Ru foil (12) (Table S3), rather speaking for the presence of large amounts of Ru suspension bonds due to the unique 2D structure as already demonstrated by the electron microscope observation (Fig. 1a–c). The monolayer feature of 2D Ru also leads to obvious stress[31] in the Ru–Ru lattice with the bond distance increased from 2.68 Å of Ru foil and np-Ru/NC to 2.77 Å.

Generally speaking, the unique 2D structure with abundant surface atoms and suspension bonds will significantly increase the surface energy of Ru metals and thus the interfacial Schottky barriers with the NC support to enhance the rectifying effect[32,33]. The calculated Bader charge of Ru (0.04) in 2D-Ru/NC is twice as much as that (0.02) in np-Ru/NC (Fig. 2a, Supplementary Fig. 15 and Supplementary Data 1 and 2), further illustrating the key role of unique 2D structure in amplifying the electron deficiency of Ru metal even on the same NC support model with fixed Ru atom number. Experimentally, the gradual shift in Ru $3p$ XPS peaks to higher energy (Fig. 2b) from 461.9 eV of the neutral Ru nanoparticle supported on carbon support (np-Ru/C) via 462.2 eV of np-Ru/NC to 462.9 eV of 2D-Ru/NC directly reveals the electron donation from the Ru metal to the NC supports and also the more pronounced electron deficiency of 2D-Ru metal as compared with Ru nanoparticles. With more electrons donated to the NC support, the 2D-Ru metals have much higher work function (8.67 eV) as compared with the Ru nanoparticle-based hybrids (Fig. 2c and Supplementary Fig. 16). To further explore the electronic structure of the 2D-Ru/NC sample, the surface potential of NC and 2D-Ru/NC samples was carefully analyzed using Kelvin probe force microscopy techniques. With similar heights (Fig. 2d, 2g), the contact potential difference ($\Delta V$) between the Si substrate and the sample decreases from 53 mV of bare NC (Fig. 2e, f) to 34 mV (Fig. 2h, i) of the 2D-Ru/NC hybrid, indicating lower surface potential of the NC support in 2D-Ru/NC sample, which is contributed by the formation of heterojunction with more electrons flowing from the 2D-Ru metals to the same NC support.

## Electrocatalytic performance of $NO_3RR$

The amplified electron deficiency of 2D Ru metals inspired us to further evaluate its electrocatalytic performance for $NO_3RR$ using additive-free nitrate solution. As depicted in Fig. 3a, bare NC showed negligible activity for $NO_3RR$ in 1 M $KNO_3$ solution. The markedly increased $NO_3RR$ current densities of the 2D-Ru/NC in 1 M $KNO_3$ as compared to the measured HER current density in 0.5 M $K_2SO_4$ (Supplementary Fig. 17) under fixed potentials directly reveal the preference of $NO_3RR$ on the 2D-Ru/NC electrocatalyst. Additionally, the crucial role of as-formed 2D Ru metal is demonstrated by the significantly enhanced current densities of 2D-Ru/NC as compared to those of the pre-catalyst a-Ru/NC (Supplementary Fig. 18). The concentration of produced ammonium was quantified using the standard calibration curve by $^1H$ NMR analysis and colorimetric method (Supplementary Fig. 19 and "Methods" section). The observed nuclear magnetic peaks of $^{15}NH_4^+$ when feeding $^{15}NO_3^-$ as the nitrogen source (Fig. 3a inset) confirm ammonia production from the conversion of nitrate rather than external N-containing pollution. Under fixed conditions, the 2D-Ru/NC catalyst achieved efficient conversions of $NO_3^-$ to $NH_3$ with a high yield (55.4 mg cm$^{-2}$ h$^{-1}$) and FE value (>99%) at −1.1 V vs. reversible hydrogen electrode (RHE) in 1 M $KNO_3$ solution, which is markedly superior to NC, np-Ru/C, and np-Ru/NC (Fig. 3b and Supplementary Fig. 20). The FE value for $NO_2^-$ by the 2D-Ru/NC is as low as -0.2%, while no gaseous products and $N_2H_4$ were detected (Supplementary Fig. 21). More importantly, the $NH_3$ yields of np-Ru/C, np-Ru/C, and 2D-Ru/NC catalysts increased gradually, matching well with the trend of gradually enhanced electron deficiency of Ru metals (Fig. 2b).

Within a wide voltage range (−0.9 to −1.4 V vs. RHE), the 2D-Ru/NC catalyst maintains high reaction rates and FE values (>92%), demonstrating an exceptionally broad voltage applicability for $NO_3RR$ (Fig. 3c and Supplementary Fig. 22). Additionally, the 2D-Ru/NC electrocatalyst could provide excellent FE values between 97-99% to $NH_3$ in a wide range of nitrate concentrations (0.001 M to 1 M, Supplementary Fig. 23) or in different nitrate solutions ($LiNO_3$, $NaNO_3$, $KNO_3$, and $CsNO_3$, Supplementary Fig. 24). Moreover, the high activity of the 2D-Ru/NC catalyst could be consistently reproduced during the following 12 circles of reuses (Fig. 3d and Supplementary Fig. 25), with well-maintained crystallinity (Supplementary Fig. 26), morphology (Supplementary Fig. 27) and chemical composition (Supplementary Fig. 28 and Supplementary Table 1). It's worth mentioning that the pH value of the cathode electrolyte becomes as high as 10.2 within 3 min and finally to around 13.0 after two hours of the potentiostatic test at −1.1 V vs. RHE (Supplementary Fig. 29).

## Reaction mechanism of $NO_3RR$ over 2D-Ru/NC catalyst

To further elucidate the key role of the electron deficiency of 2D-Ru/NC electrode in $NO_3RR$ process, in situ Fourier transform infrared spectroscopy (FTIR) was employed to monitor possible reactants and intermediates (Fig. 4a). The typical peak[34] of as-formed $NH_3$ at 1084 cm$^{-1}$ gradually increased under elevated working potentials from 0.2 to −1.4 V vs. Ag/AgCl, indicating the direct conversion of $NO_3^-$ to $NH_3$ on the electron-deficient surface of Ru under applied potentials. The signals of $NO_2^*$ (1238 cm$^{-1}$), NO* (1277 cm$^{-1}$) and $NH_2OH^*$ (1168 cm$^{-1}$)[7,35] as possible intermediates are gradually enhanced under even higher working potentials. Moreover, the hydrogenation intermediates (−NH at 1140 cm$^{-1}$ and −NH$_2$ at 1390 cm$^{-1}$)[36,37] were further detected in in situ Raman spectra (Fig. 4b).

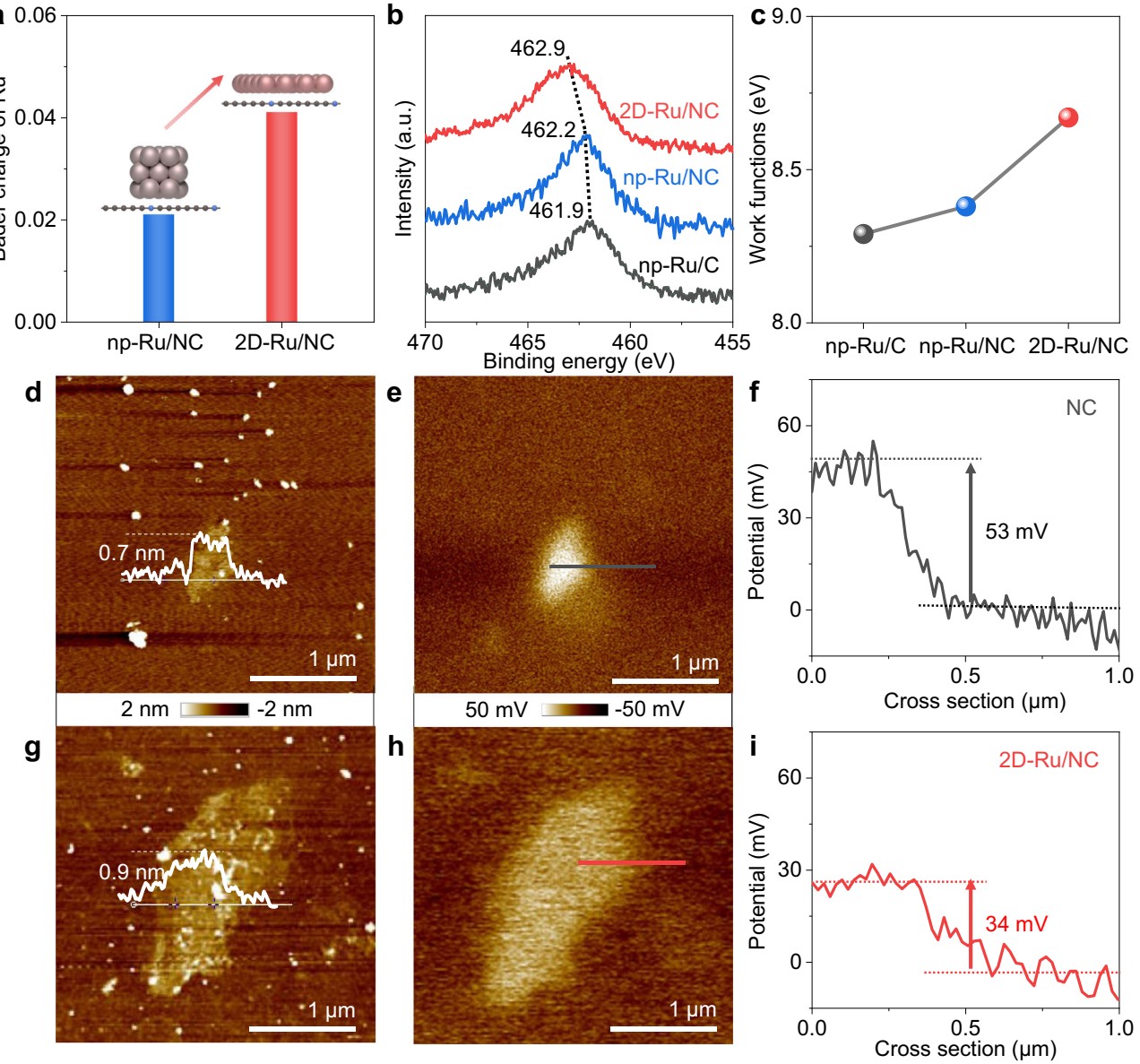

**Fig. 2 | Electric structure of the 2D-Ru/NC catalyst. a** Averaged Bader charge of Ru in 2D-Ru/NC and np-Ru/NC models. **b, c** Ru *3p* XPS spectra (**b**) and work functions (**c**) of np-Ru/C, np-Ru/NC, and 2D-Ru/NC. **d, e** AFM images (**d**) and surface electric field distribution (**e**) of NC. **f** Surface potential values extracted across the black line in (**e**). **g–i** AFM images (**g**) and surface electric field distribution (**h**) of 2D-Ru/NC. **i** Surface potential values extracted across the red line in (**h**). Source data for Fig. 2a–c, f, i are provided as a Source Data file.

Accordingly, the calculated free energies (Fig. 4c and Supplementary Table 4) using the density functional theory (DFT) of each step as revealed in the spectra analysis results (Fig. 4a, b) slightly decrease on the electron-deficient Ru-0.04 model as compared to those on the neutral Ru model. The ΔG values for the hydrogenation of the *NO to the *NOH intermediate as the rate-determining step[38] (RDS) also decrease from 0.91 eV of the neutral Ru model to 0.79 eV of the electron-deficient Ru-0.04 model, so do the calculated transition state (TS) energy barriers (Fig. 4d) from 1.32 to 1.19 eV. More importantly, the TS energy barrier of the RDS step for NO₃RR is also lower than that of the Tafel step of HER (1.42 eV, Fig. 4e) on the same Ru-0.04 model. All these results indicate the key role of the more pronounced electron deficiency of 2D Ru metal in facilitating the whole NO₃RR process but depressing HER to ensure selectivity. Indeed, the as-formed 2D-Ru/NC electrocatalyst with the most pronounced electron deficiency performs as the outstanding NO₃RR electrocatalyst with an ammonia yield of 55.4 mg cm⁻² h⁻¹ at

−1.1 V vs. RHE in 1 M KNO₃ solution and high FE (>99%), out-performing all reported electrocatalyst in neutral condition[3,38–47] (Fig. 4f and Supplementary Table 5) and Ru-based NO₃RR electrocatalysts (Table S6) in the literature. Moreover, in the more challenging pure nitrate solution without any acid, base, or other additive, the performance of 2D-Ru/NC far surpasses that of the reported catalyst[4,10] (Fig. 4f).

**Exploration on K⁺ ions spelling effect**

Besides the high activity of the electron-deficient 2D Ru centers for ampere-level NO₃RR reduction, the electron-deficient 2D Ru-based electrode also exhibits a tunable interfacial synergy of water and cations to depress the HER process and maintain satisfied FE values. By using the in situ Raman spectra (Fig. 5a and Supplementary Fig. 30), we could firstly identify the structures of interfacial water on the surface of 2D-Ru/NC electrodes[48,49], including 4-coordinated hydrogen-bonded water (4-HB·H₂O), 2-coordinated hydrogen-bonded water

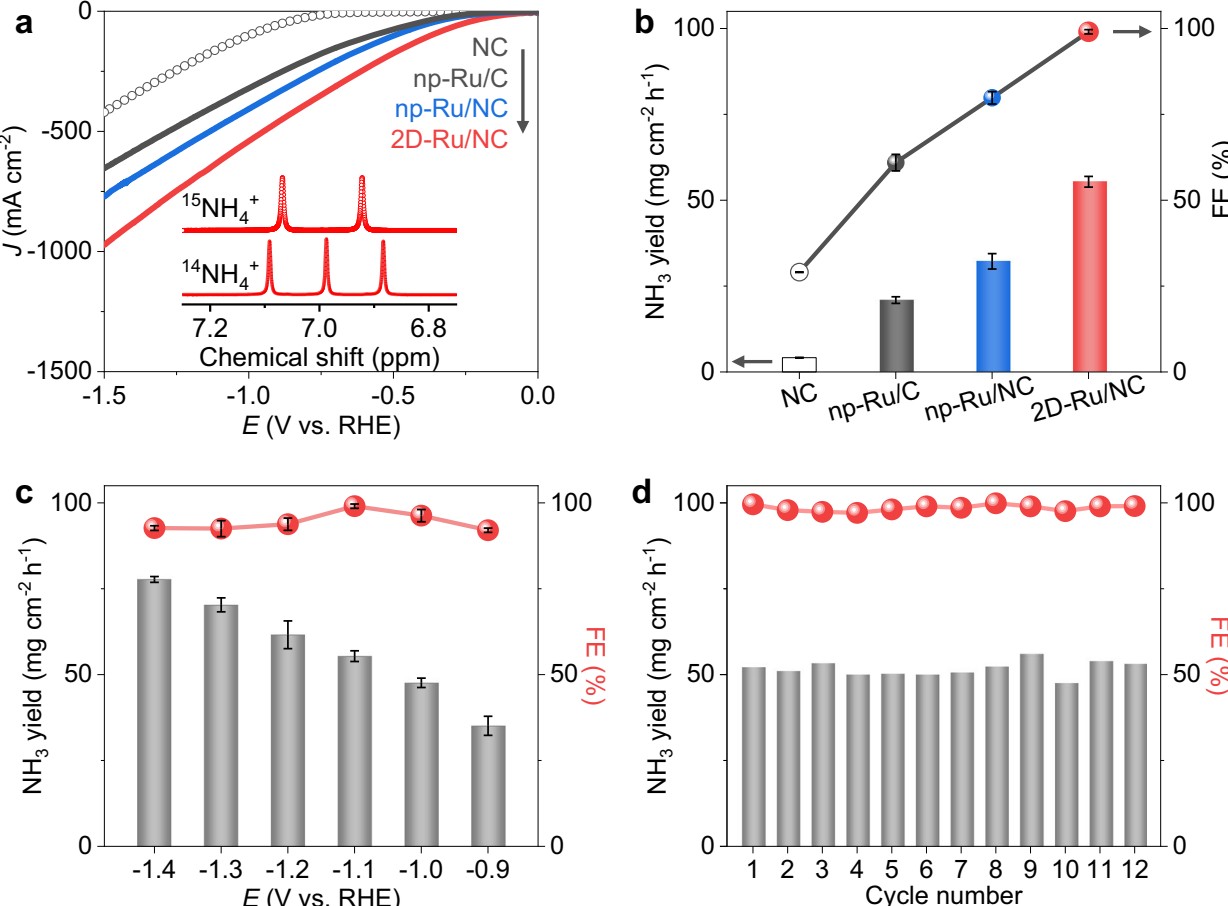

**Fig. 3 | Electrocatalytic NO₃RR performance over 2D-Ru/NC catalyst. a** LSV curves of bare NC, np-Ru/C, np-Ru/NC, and 2D-Ru/NC catalysts in 1 M KNO₃. (The solution resistances of NC, np-Ru/C, np-Ru/NC and 2D-Ru/NC are 5.47 ± 0.05 Ω, 4.71 ± 0.01Ω, 4.56 ± 0.2 Ω and 4.65 ± 0.2 Ω, respectively) Inset: ¹H NMR spectra of the resultant electrolyte using ¹⁵NO₃⁻ and ¹⁴NO₃⁻ as nitrogen source. **b** NH₃ yield rates and FE values of bare NC, np-Ru/C, 2D-Ru/NC, and np-Ru/NC catalysts at −1.1 V vs. RHE for NO₃RR. **c** NH₃ yield rates and FE values of 2D-Ru/NC catalyst under various potentials. **d** NH₃ yield rates and FE values of 2D-Ru/NC catalyst at −1.1 V vs. RHE with each cycle for 2 h. All the voltages were not IR corrected. The error bars were the relative standard deviations obtained by three repeated tests. Source data for Fig. 3 are provided as a Source Data file.

(2-HB·H₂O), and hydrated K⁺ ion water (K⁺·H₂O). The adsorption of K⁺·H₂O on both 2D-Ru/NC and np-Ru/NC electrodes with electron-deficient Ru surface is not sensitive to the applied potentials (Fig. 5b and Supplementary Table 7), indicating their weak interaction with electrode surfaces. Furthermore, the vibrational frequency of K⁺·H₂O on 2D-Ru/NC (3618 cm⁻¹) is higher than that of np-Ru/NC (3590 cm⁻¹) at −1.4 V vs. Ag/AgCl (Supplementary Fig. 31), suggesting a weaker interaction of K⁺·H₂O groups with 2D-Ru/NC electrode[48] and thus a more pronounced repelling effect as also demonstrated by a much lower proportion (4.8%) of K⁺·H₂O in the electric double layer as compared with the proportion (14.9%) of np-Ru/NC electrode (Fig. 5b). Indeed, AIMD simulation results (Fig. 5c, Supplementary Figs. 32 and 33 and Supplementary Data 3 and 4) directly exhibit the selective repelling effect of the electron-deficient Ru surface on K⁺ cations with a fast departure of hydrated K⁺ ions away from the electrode surface within 0.4 ps. It should be noted that all anions including NO₃⁻ and OH⁻ ions are not repelled away by the electron-deficient Ru surface (Fig. 5c, d and Supplementary Figs. 34 and 35) but are enriched[3,50], which further benefits the NO₃RR process with gradually promoted current output (Supplementary Fig. 29) for the following reduction process. Moreover, the AIMD simulation results (Supplementary Fig. 36) indicate that the unique cation repelling effect of electron-deficient Ru surface is also available for Li⁺, Na⁺ and Cs⁺ ions

with relatively lower NH₃ yields as compared to K⁺ ions (Supplementary Fig. 24).

More importantly, such a unique repelling effect on hydrated K⁺ ion is dominated by the electron deficiency of Ru-based electrodes according to simulation results on the pristine Ru model and electron-deficient Ru models (Ru-0.02 and Ru-0.04) (Fig. 5d, e). The average distance of K⁺ ions from the Ru surface gradually increased from 6.6 Å of Ru model via 7.1 Å of Ru-0.02 model to 8.2 Å of Ru-0.04 model (Fig. 5d and Supplementary Fig. 33), matching well with the trend of electron-density dependent repelling effect of hydrated K⁺ ions revealed by the proportion and vibrational frequency of K⁺·H₂O group in Raman spectra (Fig. 5b and Supplementary Fig. 31). Accordingly, the water density near the electron-deficient Ru electrode surface significantly decreases (Fig. 5e) with more water molecules taken away by the repelled K⁺ ions. Experimentally, the proportion of 4-HB·H₂O (Fig. 5b) on the 2D-Ru/NC electrode surface always stays lower than that on the np-Ru/NC electrode, highlighting the crucial role of electron deficiency of Ru in loosening the water layer to depress HER[51,52] and accelerate the nitrate penetration.

## Ampere-level NO₃RR experiments
Taken together, the electron-deficient Ru repels the hydrated K⁺ ions away from the surface and decreases the density of interfacial water,

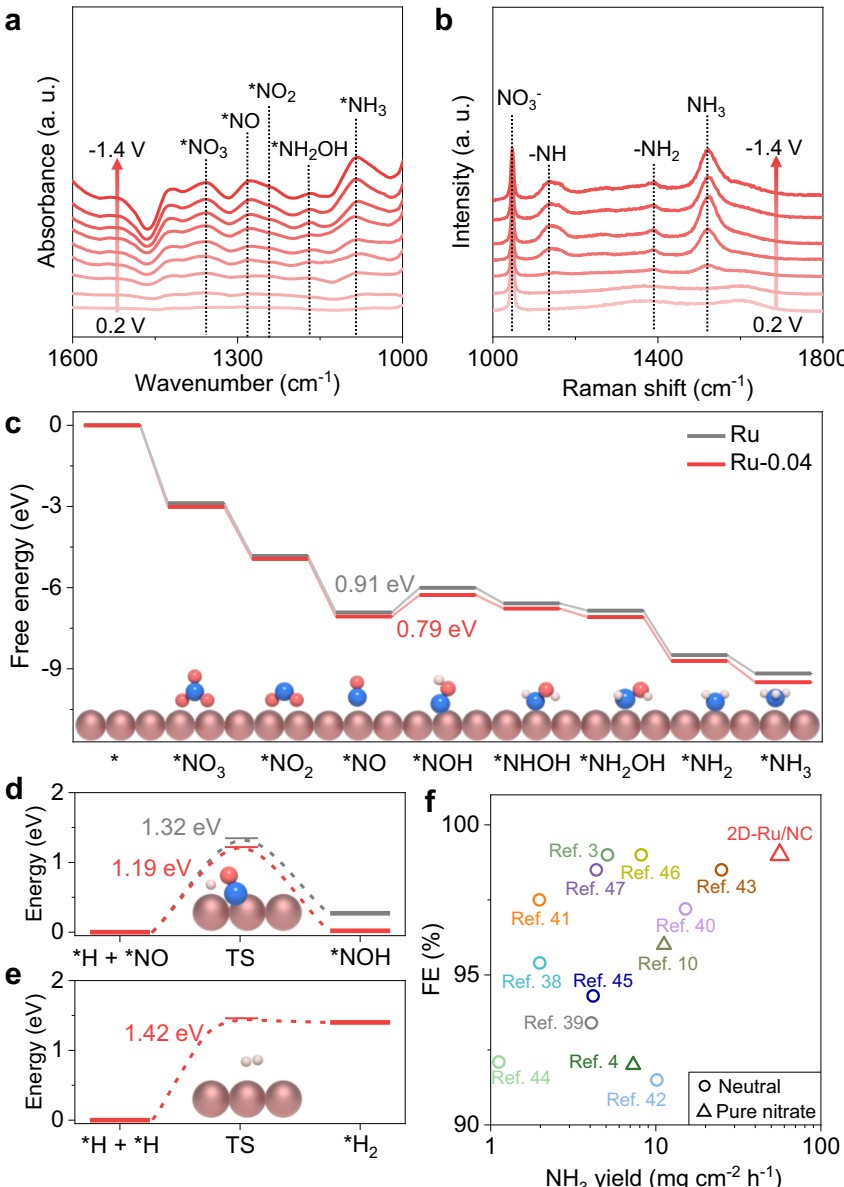

**Fig. 4 | Reaction mechanism for NO₃RR over 2D-Ru/NC catalyst. a** In situ FTIR spectra of 2D-Ru/NC in 1 M KNO₃ under various potentials. **b** In situ Raman spectra of 2D-Ru/NC in 1 M KNO₃ under various potentials. **c** Gibbs free energy diagram of NO₃RR process on Ru (black) and Ru-0.04 models (red). Insets: the configurations of each step on Ru-0.04 model (Ru, Red brown; O, red; N, blue; H, white). **d** Energy barriers for hydrogenation of the *NO on Ru (black line) and Ru-0.04 models (red line). Inset: the configuration of the TS of the hydrogenation step on Ru-0.04 model. **e** Energy barrier for the TS of the Tafel step of HER on Ru-0.04 model. Inset: the configuration of the transition state on Ru-0.04 model. **f** NO₃RR performance of 2D-Ru/NC in pure nitrate solution as compared with bench-marked electrocatalysts in pure nitrate solution (triangles) or neutral solution with supporting electrolyte (circles). Source data for Fig. 4a–e are provided as a Source Data file.

which will depress the HER (Supplementary Fig. 37) but accelerate the reloading of nitrate on the Ru surface (Fig. 6a). In this way, the as-formed 2D-Ru/NC catalyst with pronounced electron deficiency can help to maintain high FE values for ampere-level NO₃RR of pure nitrate solution for long-term uses. Exactly, the 2D-Ru/NC catalyst exhibited a lower working voltage than np-Ru/NC catalyst (Fig. 6b) in pure solution of 1 M KNO₃ at 1 A cm⁻², experimentally validating the largely reduced energy barriers for nitrate reduction (Fig. 4c, d) by the more pronounced electron deficiency of 2D-Ru-based electrode surface.

The obvious decline in yield and FE are the general challenge for long-term uses in the literature and also in our np-Ru/NC cathode-based reactor (blue line and bars of Fig. 6c). The yield rate of ammonia decreased from 55 mg mg$_{cat}^{-1}$ h⁻¹ to 39 mg mg$_{cat}^{-1}$ h⁻¹ within 6 h in our np-Ru/NC cathode-based reactor with the FE value decreasing from 70% to 49%. Unexpectedly, the 2D-Ru/NC cathode-based reactor (red

line and bars of Fig. 6c) could provide consistent ammonia yield rates with well-maintained FE values (90–94%), producing 0.26 M ammonia solution via six-hour ampere-level reduction without any concentration and purification process, which is 8.1 times the highest concentration[9] reported till now (Fig. 6d). Moreover, the 2D-Ru/NC cathode-based reactor can work effectively and durably in concentrated nitrate solutions without any additives, giving a high FE (94%) and NH₃ yield (74.8 mg mg$_{cat}^{-1}$ h⁻¹) as compared with the reported Ru-based electrocatalysts under ampere-lever current density[9,53]. To further demonstrate the stability of the 2D-Ru/NC electrode, a long-term stability test for more than 120 h was conducted under ampere-level current density (Supplementary Fig. 38). The well-maintained working voltage without obvious Ru leaching (Supplementary Table 1) again speaks for the good stability of the 2D-Ru/NC electrode even at ampere level. All these results indicate the potential

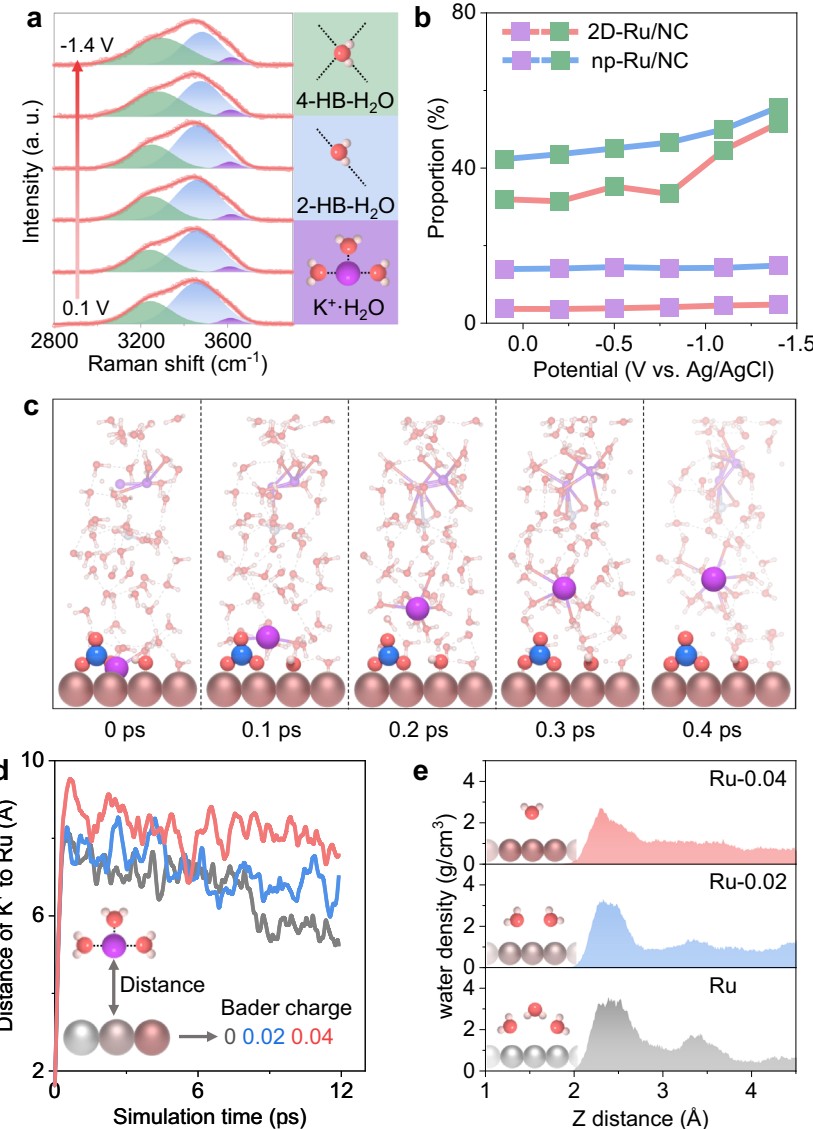

**Fig. 5 | Boosted K⁺ ions repelling effect over 2D-Ru/NC electrode. a** Potential-dependent in situ Raman spectra of interfacial water on 2D-Ru/NC electrode surface during NO₃RR process. Gaussian fits of three O−H stretching models of 4-HB·H₂O, 2-HB·H₂O, and K⁺·H₂O are shown in green, blue, and purple, with corresponding schematic structures on the right side. **b** Estimated proportions of 4-HB·H₂O (green squares) and K·H₂O (purple squares) from the peak areas of **a** on 2D-Ru/NC (red lines) and np-Ru/NC (blue lines) surface. **c** Snapshots of pre-adsorbed various ions on the Ru-0.04 model surface at 0, 0.1, 0.2, 0.3, and 0.4 ps using AIMD method. Ru, red brown; O, red; N, blue; K, purple; H, white. **d** Distance of pre-adsorbed K⁺ in **c** to the surface of neutral Ru (black line), Ru-0.02 (blue line), and Ru-0.04 (red line) models. **e** Distributions of simulated water density along the Z direction from 1 Å to 4.5 Å. Ru-0.02 model for np-Ru/NC sample and Ru-0.04 model for 2D-Ru/NC sample based on calculated Bader charges (Fig. 2a). Source data for Fig. 5b, d, e are provided as a Source Data file.

of 2D-Ru/NC sample as a powerful and durable catalyst for large-scale electrocatalytic ammonia synthesis and energy storage.

## Discussion

In summary, we presented the conceptual application of pronounced electron deficiency of a unique 2D Ru metal structure to boost the K⁺ ions repelling effect to maintain the high yields and FE values of ammonia production from long-term ampere-level nitrate reduction reaction. The pronounced electron deficiency dominates the unique repelling effect on the hydrated K⁺ ions, as indicated by the distance of K⁺ ions to the catalyst surface, to loosen the water layer, accelerate the penetration of NO₃⁻ ions and depress the hydrogen evolution. The 2D-Ru/NC sample achieved ampere-level NO₃RR reduction in additive-free nitrate solution with high yields and well-maintained FE values,

meeting the demands of large-scale electrocatalytic ammonia synthesis and energy storage. Moreover, given the exploration in the complex electrocatalytic interface, we strongly believe the detailed study on the electron-density dependent K⁺ ions repelling effect for regulating the distribution of specific cations can deepen the understanding of electrocatalytic processes in the electric double layer and accelerate the design and synthesis of electrocatalysts for the sustainable conversions of a vast series of small molecules into valuable products.

## Methods

### Chemicals and materials

All chemicals were purchased from commercial sources without further purification. Urea (CH₄N₂O, 99.5 %, Aladdin), 1,

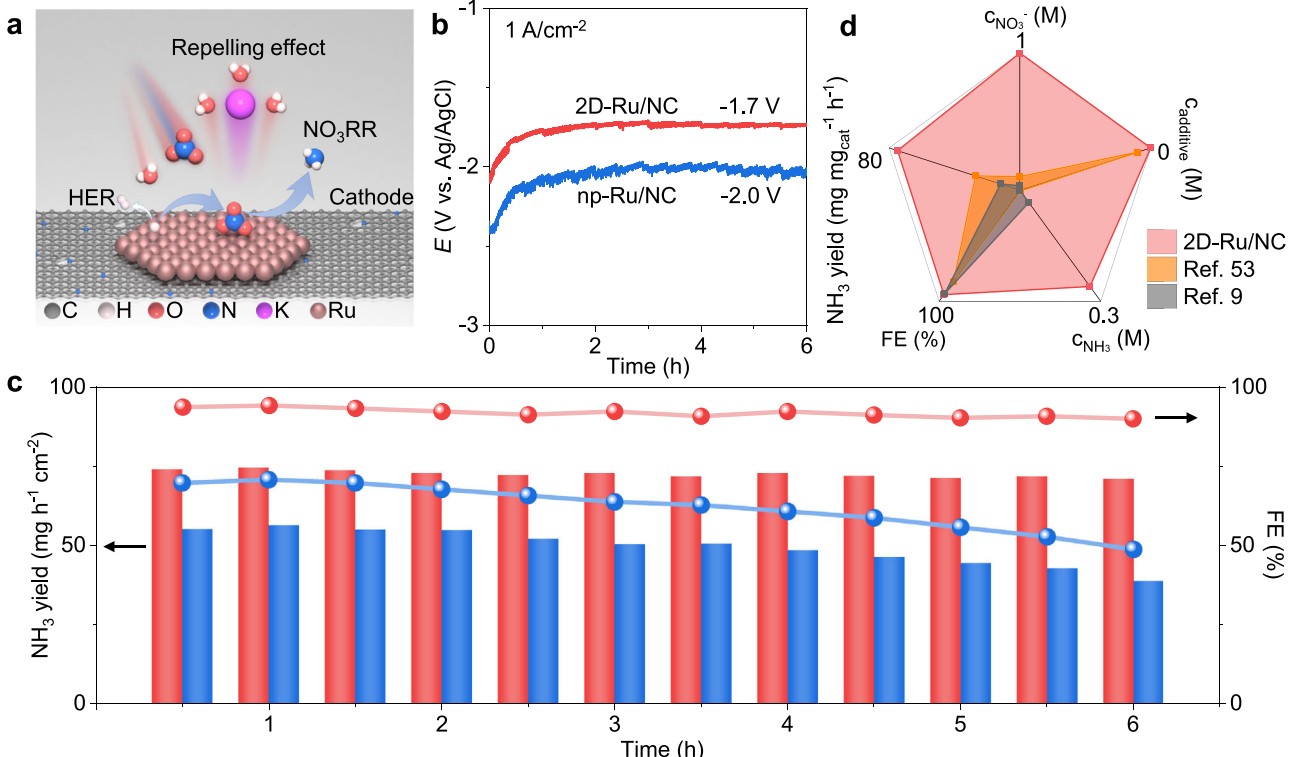

**Fig. 6 | Superiority of 2D-Ru/NC electrode for ampere-level NO₃RR. a** Scheme of hydrated K⁺ ions repelling effect for depressed HER and boosted NO₃RR. **b** Chronopotentiometry curves of NO₃RR over 2D-Ru/NC and np-Ru/NC electrodes at 1 A cm⁻² in 1 M KNO₃ solution. **c** NH₃ yields and FE values of 2D-Ru/NC (red) and np-Ru/NC (blue) electrodes during ampere-level NO₃RR process in (**b**). **d** Comparison of NO₃RR performance under ampere-lever current density over 2D-Ru/NC electrode and other reported Ru-based electrocatalysts. The five axes: nitrate concentrations in initial reaction solutions ($C_{NO_3^-}$, scale: 0 to 1 M), additive concentrations in initial reaction solutions ($C_{additive}$, scale: 1 to 0 M), ammonia concentrations of obtained solutions ($C_{NH_3}$, scale: 0 to 0.3 M), FE values (scale: 0 to 100%), NH₃ yields (scale: 0 to 80 mg $mg_{cat}^{-1}$ h⁻¹). Source data for Fig. 6b–d are provided as a Source Data file.

4-benzenedicarboxylic acid (C₈H₆O₄, 99 %, Aladdin), triethylenediamine (C₆H₁₂N₂, 98 %, Aladdin), N, N-Dimethylformamide (C₃H₇NO, 99.9 %, Aladdin), Ruthenium chloride trihydrate (RuCl₃·3H₂O, 99 %, Aladdin), Sulfuric acid (H₂SO₄, 96 %, Adamas), Nafion PFSA polymer dispersions (5 wt.%), Potassium sulfate (K₂SO₄, 99.9%, Sigma-Aldrich), Potassium nitrate (KNO₃, 99%, Sigma-Aldrich), Seignette salt (C₄H₄KNaO₆, 1.5 M in water Macklin), DMSO-d₆ (C₂SOD₆, 99.9%, Adamas), Maleic acid (C₄H₄O₄, 99 %, TCI), Carbon cloth (WOS1011, CeTech Co., Ltd), All gas (99.999%) were purchased from Air Liquide.

### Synthesis of the NC support

Typically, urea was placed into a covered crucible, heated at 550 °C for 4 h with a heat rate of 2.3 °C min⁻¹ under air atmosphere and cooled down to ambient temperature to obtain yellow powder (g-C₃N₄). 2.72 g 1,4-benzenedicarboxylic acid, and 1.92 g triethylenediamine were mixed into 150 mL N, N-Dimethylformamide, then 18.56 g g-C₃N₄ was added and dried at 120 °C under stirring. The obtained solid powder was heated at 1000 °C for 1 h with a heat rate of 1.5 °C min⁻¹ under N₂ atmosphere and cooled down to ambient temperature to obtain black powder.

### Synthesis of the a-Ru/NC catalyst

Typically, 100 mg NC support was dispersed in 10 mL of RuCl₃·3H₂O aqueous solution (20 mM). Then, the suspension was stirred and protected from the light for 24 h. The obtained pre-catalyst (denoted as a-Ru/NC) was washed thoroughly with deionized water and dried in a freeze dryer.

### Synthesis of the 2D-Ru/NC catalyst

The 2D-Ru/NC catalyst was prepared during the in situ electrochemical reduction process. For the preparation of a-Ru/NC-based work electrode, 5 mg of the a-Ru/NC pre-catalyst was dispersed in 1400 μL of ethanol, 700 μL of deionized water, and 100 μL of 10 wt% Nafion aqueous solution by sonicating for 2 h to get a homogeneous catalyst ink. Then, a certain volume of the ink was dropped onto a carbon cloth and dried at 60 °C for 2 h. The electrochemical reduction of a-Ru/NC pre-catalyst was performed in a H-type cell with a Nafion 117 proton exchange membrane on a CHI 660E electrochemical analyzer (CH Instruments, Shanghai) in the Ar-saturated 0.5 M H₂SO₄ electrolyte. The Nafion 117 proton exchange membranes (thickness: 183 μm) were pretreated separately in hydrogen peroxide (5 wt%), deionized water, and H₂SO₄ (5 wt%) for one hour at 80 °C. Then the membranes were immersed in deionized water for later use. The Ag/AgCl electrode and Pt mesh (1 × 1 cm) were used as the reference electrode and counter electrode, respectively. The continuous LSV scanning were performed at a sweep rate of 10 mV s⁻¹ and the potential range of 0.5 ~ −0.5 V versus RHE until the current density reached steady state to obtain 2D-Ru/NC catalyst for further characterizations and electrocatalytic reactions.

### Synthesis of the np-Ru/NC catalyst

100 mg NC support was firstly dispersed in 20 mL of deionized water and ultrasonic mixing for 10 min to obtain a homogenous black suspension, then 2 mL of 20 mM RuCl₃·3H₂O aqueous solution was added slowly. The suspension was stirred and protected from the light for

12 h. After that, the pH value was adjusted to around 12 with 1 M NaOH aqueous solution with stirring for 4 h. Then, 2 mL of 1 M NaBH$_4$ aqueous solution was added slowly with stirring for 4 h. The obtained sample was washed thoroughly with deionized water and dried in a freeze dryer. The black sample was heated at 500 °C for 2 h with a heat rate of 5.0 °C min$^{-1}$ under 5% H$_2$/Ar atmosphere and cooled down to ambient temperature to obtain the np-Ru/NC catalyst.

## Synthesis of the np-Ru/C catalyst

The synthesis method of np-Ru/C catalyst is consistent with that of np-Ru/NC catalyst, except for replacing the NC support with Carbon black (vxc72, treatment at 1000 °C under N$_2$ atmosphere).

## Characterizations

Aberration-corrected environmental scanning transmission electron microscope dark field (ESTEM-DF) images and secondary electron (ESTEM-SE) images were recorded on a Hitachi HF5000 microscope at 200 kV. Powder X-ray diffraction (XRD) patterns were performed on a MiniFlex600 X-ray diffractometer with Cu Kα radiation ($\lambda = 1.5418$ Å) and operated at a scan rate of 1° min$^{-1}$. X-ray photoelectron spectroscopy (XPS) experiments were performed using a ThermoFisher ESX-CALAB Xi$^+$ spectrometer with monochromatized Al Kα radiation and we conducted XPS test using catalyst-based electrodes as the experimental samples. Ultraviolet photoelectron spectroscopy (UPS) experiments were performed using a Kratos Axis Ultra DLD spectrometer and we conducted UPS test using catalysts-based electrodes as the experimental samples. Inductively coupled plasma–atomic emission spectroscopy (ICP-AES) measurements were conducted on a PerkinElmer Optima 3300DV spectrometer for metal contents analysis. X-ray absorption fine structure (XAFS) spectra at the Ru K-edge of all samples were recorded in transmission or fluorescence mode at room temperature at the BL14W1 beamline of the Shanghai Synchrotron Radiation Facility (SSRF), the X-ray energy was monochromatized with a Si (311) double crystal monochromator. The position of the absorption edge ($E_0$) was calibrated using standard Ru foil at ambient conditions. The measured EXAFS data were processed according to standard procedures using the ATHENA program, and curve fitting analysis of EXAFS data was carried out using the ARTEMIS program in the IFEFFIT software packages. The surface potential and height images were measured using Kelvin probe force microscopy techniques (Dimension XR) under an ambient atmosphere.

## Electrocatalytic NO$_3^-$ reduction reaction (NO$_3$RR)

The NO$_3$RR performance were measured in 1 M KNO$_3$ solution. 101.1 g KNO$_3$ was dissolved in deionized water and diluted to 1 L using a 1 L volumetric flask. Then, the 1 M KNO$_3$ electrolyte was stored in a plastic bottle away from light. The NO$_3$RR measurement was performed in a H-type cell with a Nafion 117 proton exchange membrane. The obtained 2D-Ru/NC electrode (0.5 × 0.5 cm, loading: 1.0 mg cm$^{-2}$) worked as the working electrode. The Ag/AgCl electrode and Pt mesh (1 × 1 cm) were used as the reference electrode and counter electrode. The reference electrode was converted to RHE according to the Nernst equation ($E_{RHE} = E_{Ag/AgCl} + 0.059 \times pH + \varphi_{Ag/AgCl}$). The $\varphi_{Ag/AgCl}$ term was calibrated by a hydrogen reversible reaction (Supplementary Fig. 39). The presented current density was normalized to the geometrical area (0.25 cm$^{-1}$) of the work electrode. All the polarization curves were the steady-state curves obtained after several sweep cycles at a scan rate of 10 mV s$^{-1}$. A potentiostatic test was conducted in an Ar-saturated 1.0 M KNO$_3$ aqueous solution for 1 h at a stirring rate of 350 rpm. The solution resistances were measured using AC-impendence test from 0.1 Hz to 100 kHz with a voltage perturbation of 5 mV. All the voltages in this work were not IR corrected. After electrolysis, the products in the electrolyte were measured using the following methods, respectively. All the electrochemical experiments were carried out at room temperature.

## Quantification of ammonia (NH$_3$)

**$^1$H NMR method.** The NH$_3$ concentration in the electrolyte was determined by $^1$H nuclear magnetic resonance (Bruker Avance III HD NMR, 500 MHz) with using DMSO-$d_6$ as the solvent and maleic acid as the internal standard. The calibration curve was made as follows. First, a series of NH$_4$Cl aqueous solutions with known concentration were prepared and the pH value was adjusted to around 2.0 with HCl solution. Second, 0.4 mL of the NH$_4$Cl aqueous solution was mixed with 0.2 mL DMSO-$d_6$ (with 3 mg mL$^{-1}$ maleic acid) for NMR test at room temperature. Finally, the calibration curve was achieved using the peak area ratio between NH$_4^+$ and maleic acid. For testing the produced ammonia after NO$_3$RR, the electrolyte was taken out from the electrolytic cell and the pH value was adjusted to 2.0 with HCl solution. Then, the process is the same to that for making the calibration curve. The ammonia concentration can be calculated from the peak area ratio using the calibration curve.

**Colorimetric method.** The NH$_3$ concentration was also determined by the Nessler method. First, a certain amount of electrolyte was taken out from the electrolytic cell and diluted to the detection range. Then, 5 ml of the diluted electrolyte was removed, and 0.5 mL of 0.2 M C$_4$H$_4$KNaO$_6$ aqueous solution was added, followed by the addition of 0.5 mL of Nessler solution. After standing at room temperature for 20 min, the UV-vis absorption spectrum was measured at a wavelength of 420 nm (Scanning range: 600–400 nm) (SHIMADZU UV-2450 UV-vis spectrophotometer). The concentration of NH$_3$ was determined via a calibration curve, which was prepared using a series of NH$_4$Cl aqueous solutions with known concentrations.

**Quantification of NO$_2^-$.** The NO$_2^-$ concentration was determined using the Griess method. The color reagent was prepared by mixing p-aminobenzenesulfonamide (4.0 g), N-(1-Naphthyl)ethylenediamine dihydrochloride (0.2 g), phosphoric acid (10 mL), and deionized water (50 mL). After electrocatalytic reaction, a certain amount of electrolyte was taken out and diluted to the detection range. Then, 0.1 mL of the color reagent was added into the 5 mL of the diluted electrolyte. After standing at room temperature for 20 min, the UV-vis absorption intensity was measured at a wavelength of 540 nm. The concentration of NO$_2^-$ was determined via a calibration curve, which was prepared using a series of KNO$_2$ aqueous solutions with known concentrations.

**Quantification of N$_2$H$_4$.** The presence of N$_2$H$_4$ in the electrolytes was determined using the Watt-Chrisp method. The color reagent was prepared by mixing p-(dimethylamino)benzaldehyde (2.0 g), 1.0 M HCl (20 mL) and ethanol (100 mL). After electrocatalytic reaction, 5 mL electrolyte was taken out and 5 ml of 1.0 M HCl was added. Then, 5 mL of the color reagent was added into the solution. After standing at room temperature for 20 min, the UV-vis absorption intensity was measured at a wavelength of 460 nm. The calibration curve was constructed by using a series of N$_2$H$_4$ aqueous solutions with known concentrations.

**Determination of H$_2$ and N$_2$.** H$_2$ and N$_2$ was qualified using gas chromatograph (GC-2014 SHIMADZU) with molesieve 13X packed column. Argon gas was used as carrier gas. The calibration curve was constructed by injecting different volumes of H$_2$ or N$_2$.

**Calculation of the Faradaic efficiency (FE) and yield rate for NO$_3$RR.**

$$\text{FE}_{\text{NH}_3} = (8 \times F \times C_{\text{NH}_3} \times V \times 10^{-3})/Q \tag{1}$$

$$\text{Yield}_{\text{NH}_3} = [(C_{\text{NH}_3} \times V \times M_{\text{NH}_3})/(t \times A)]\text{mg cm}^{-2}\,\text{h}^{-1} \tag{2}$$

$$\text{FE}_{\text{NO}_2^-} = (2 \times F \times C_{\text{NO}_2^-} \times V \times 10^{-3})/Q \qquad (3)$$

$$\text{Yield}_{\text{NO}_2^-} = [(C_{\text{NO}_2^-} \times V \times M_{\text{NO}_2^-})/(t \times A)]\text{mg cm}^{-2}\,\text{h}^{-1} \qquad (4)$$

Where $F$ is the Faraday constant (96485 C mol$^{-1}$), $C_{\text{NH}_3}$ is the measured $NH_3$ concentration (mM), $V$ is the volume of electrolyte (0.025 L), $Q$ is the total charge passing the electrode (C), $M_{\text{NH}_3}$ is the molar mass of NH3 (17 g mol$^{-1}$), $M_{\text{NO}_2^-}$ is the molar mass of $NH_3$ (46 g mol$^{-1}$), t is the electrolysis time, $A$ is the geometric area of working electrode (0.25 cm$^2$).

**$^{15}$N isotope-labeling experiment.** An isotope-labeling experiment using K$^{15}$NO$_3$ (99 atom % $^{15}$N) aqueous solutions was carried out to clarify the source of ammonia. After 1 h of electrocatalytic reduction at −1.1 V versus RHE, the obtained $^{15}$NH$_4^+$ was tested by $^1$H NMR method.

**Measurement of pH value.** The pH values of the solution were measured using a SX-620 meter (SANXIN, Shanghai). Before measurement, the pH meter was calibrated with three buffer solutions (pH = 4.00, 7.00, and 10.01). The pH value of the target solution was measured three times.

**Electrochemical in situ FRIT measurement.** The in situ FTIR measurements were carried out on a Nicolet iS50 FTIR spectrometer (Supplementary Fig. 40a). A H-type three-electrode cell was used for FTIR experiments, with platinum wire as counter electrode and Ag/AgCl electrode as reference electrode. A carbon cloth loaded with catalyst (loading: 1 mg cm$^{-2}$) was used as a working electrode. 1 M KNO$_3$ solution was used as electrolyte. All spectra were shown in the absorbance unit as $-\log(I/I_0)$, where $I$ and $I_0$ represent the intensities of the reflected radiation for sample and reference single-beam spectra.

**Electrochemical in situ Raman measurement.** The in situ Raman measurements were carried out by Renishaw inVia Qontor Raman microscope. The excitation wavelength of the semiconductor laser was 532 nm, and the laser power was 2.5 mW. All the Raman measurements were performed with a 50× microscope objective. Raman frequencies were calibrated using silicon wafers before experiment. A H-type three-electrode Raman cell with an embedded quartz window was used for Raman experiments (Supplementary Fig. 40b), with platinum wire as counter electrode and Ag/AgCl electrode as reference electrode. A carbon cloth loaded with catalyst (loading: 1 mg cm$^{-2}$) was used as a working electrode. 1 M KNO$_3$ solution was used as electrolyte. Before Raman test, the working electrode underwent a potentiostatic test to ensure the hydrophilicity. The Vertex.C potentiostat was used to control the potential during the Raman test.

**Density functional theory (DFT) calculations.** DFT calculations were performed using the Vienna Ab Initio Simulation Package (VASP)[54,55]. The generalized gradient approximation (GGA) with the Perdew−Burke−Ernzerh (PBE) of exchange correlation functional[56] was used. For the plane-wave expansion, a 450 eV kinetic energy cutoff was used. The convergence criteria for energy and force were set as $10^{-5}$ eV and 0.02 eV Å$^{-1}$, respectively. The K-points were set to be $3 \times 3 \times 1$ for geometry optimization and electronic structure analysis. The van der Waals dispersion correction described by the DFT-D3 approach[57,58] was taken into consideration during the calculations. The transitional state searches were conducted using the climbing image nudged elastic band (CI-NEB) method[59].

For the Bader charge analysis between Ru metal and NC, a $7 \times 7$ N-doped graphene sheet was built according to the C/N ratio and the concentration of graphitic/pyridinic N dopants by the XPS results (Supplementary Fig. 8). In order to obtain more accurate differences in

electronic density caused by the 2D structure, we thus constructed a single-atom-layered Ru sheet model for the 2D-Ru/NC sample and a Ru nanoparticle model for np-Ru/NC sample with the same Ru atom numbers (Supplementary Fig. 14).

For the free energy and translation state calculation, a single-layered Ru slab was built (Ru model). According to the Bader charge analysis results (Fig. 2a and Supplementary Fig. 14), we discharged 0.04 e$^-$ per Ru atom to construct electron-deficient models of 2D-Ru/NC sample (Ru-0.04 model). The Gibbs free energy change ($\Delta G$) for each step was calculated as: $\Delta G = \Delta E + \Delta \text{ZPE} - T\Delta S$ ($T = 298.15$ K), where $\Delta E$, $\Delta \text{ZPE}$, and $\Delta S$ are the changes in the reaction energy, zero-point energy, and entropy, respectively.

**Ab initio molecular dynamic (AIMD) simulation.** AIMD simulations were also performed via VASP. The canonical ensemble condition (NVT) was imposed by a Nose-Hoover thermostat with a target temperature of 300 K during the simulations. The time step is 1.0 fs and only the gamma point of Brillouin zone is used for all AIMD simulations. The van der Waals dispersion correction described by the DFT-D3 approach was taken into consideration during the simulations.

The AIMD simulations were carried out to study the behaviors of water molecules and ions at Ru/water interface, considering the dynamical feature of solvents with the hydrogen bonding network. The Ru/water interface was modeled by adding 54 water molecules above a single-layered $4 \times 4$ Ru slab (estimated from the water density of 1 g cm$^{-3}$ at room temperature). According to the Bader charge analysis results (Fig. 2a and Supplementary Fig. 14), We discharged 0.02 e$^-$ and 0.04 e$^-$ per Ru atom to construct electron-deficient models of np-Ru/NC sample (Ru-0.02 model) and 2D-Ru/NC sample (Ru-0.04 model), respectively. Since the electrolyte rapidly becomes alkaline during the NO$_3$RR process (Supplementary Fig. 29), hydroxide ions should be considered in the simulation. Specifically, to study the synergy of water molecules and ions, three K$^+$ ions, one OH$^-$ ion, and two NO$_3^-$ ions were added to the simulation model. In the initial model, one NO$_3^-$ ion, one OH$^-$ ion, and one K$^+$ ion were placed on the Ru surface. Then, all the periodic models were simulated for at least 12 ps. The first 2 ps of simulation are used for pre-equilibration. The statistics results of the simulation came from the data obtained from the last 10 ps of simulations. When investigating the behavior of Li$^+$, Na$^+$, and Cs$^+$ ions, the same model was used, only replacing the alkali metal ions.

## Data availability
All data that support the findings of this study are present in the paper and Supplementary Information. Further information can be acquired from the corresponding authors upon request. Source data are provided with this paper Source data are provided with this paper.

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

## Acknowledgements

This work was supported by the National Natural Science Foundation of China (22071146 for X.-H.L., and 21931005 for J.-S.C.), Shanghai Science and Technology Committee (23XD1421800 for X.-H.L.), Shanghai Shuguang Program (21SG12 for X.-H.L.), and Shanghai Municipal Science and Technology Major Project. The authors thank the Shanghai Synchrotron Radiation Facility for providing beam time (BL14W1).

## Author contributions

X.-H.L., S.-N.Z., and P.G. designed the experiments. S.-N.Z and P.G. planned and performed catalyst synthesis, conducted the performance tests and analyzed data. S.-N. Z. finished the theoretical calculation. Q.-Y.L., Z.Z., and B.-L.L. helped to conduct the experiment and characterizations. X.-H.L., S.-N.Z., and P.G. co-wrote the original paper. X.-H.L., and J.-S.C. oversaw all of the research phases. All of the authors discussed the results and commented on the paper.

## Competing interests

The authors declare no competing interests.
