## [Transparent Peer Review file · Nature Communications]

Ampere-level reduction of pure nitrate by electron-deficient Ru with K⁺ ions repelling effect

Corresponding Author: Professor Xin-Hao Li

Version 0:

Reviewer comments:

Reviewer #1

(Remarks to the Author)

This manuscript focuses on a critical challenge in ammonia synthesis via electrochemical nitrate reduction reaction powered by renewable electricity. The authors reported an important and interesting K⁺ ions repelling effect on electron-deficient Ru surface, which is totally different from the previously reported cation effects among different cations. The authors synthesized electron-deficient Ru catalysts with unique 2D Ru metal structure, which have been well characterized and analyzed. In this way, the as-formed 2D-Ru/NC catalyst with pronounced electron deficiency demonstrated excellent NO₃RR performance, especially in additive-free nitrate solution, surpassing that of the reported catalyst in literatures. To prove the crucial role of the repelling effect, detailed in situ spectra and theoretical calculation results gave in-depth mechanistic clarity that the electron-deficient Ru repels the hydrated K⁺ ions away from the surface and decreases the density of interfacial water, which would depress hydrogen evolution but promote NO₃RR. Overall, the approach of turning the electron deficiency to modify interfacial water via K⁺ ion repelling effect is highly innovative and inspiring, providing valuable insights for future catalyst design in electrocatalysis.

In general, I believe this is an excellent work. To facilitate publication, the manuscript should undergo minor revision to address the following points.

1. The in situ Raman spectra are important evidences for mechanistic clarity. The instrument parameters and experimental procedures about spectra test should be more detailed. Please provide more information.
2. To prove the stability of synthesized catalyst, the author only provided the Ru 3p XPS spectra of used catalyst. The stability of NC support is also crucial for whole catalyst. Please provide the XPS result of the NC support to confirm its stability.
3. The reported 2D-Ru/NC catalyst was synthesized by in situ electrochemical reduction method. To highlight the importance of as-formed 2D Ru structure and enhanced integrity and credibility of this work, the NO₃RR performance of an a-Ru/NC electrode should also be tested for comparison.
4. What are the key factors that contribute to the well-maintained Faradic efficiency values and yields of 2D-Ru/NC catalyst under ampere-level current density? Please provide a more detailed explanation.

Reviewer #2

(Remarks to the Author)

In this paper, the authors present an important and attractive breakthrough in electrocatalytic nitrate reduction (NO₃RR) for ammonia synthesis as a viable alternative to traditional Haber-Bosch processes. The reported 2D-Ru/NC catalyst achieves ampere-level NO₃RR reduction in additive-free nitrate solution with record-high yields and well-maintained Faradic efficiency. Both the characterization of the materials and the electrocatalytic performance tests are carried out properly and rigorously. Based on the electronic structures and catalytic performance of catalysts, as well as various characterizations and ab initio molecular dynamics results, the authors proposed K⁺ ion repelling effect to explain the enhanced NO₃RR performance. The key innovation of this work lies in the detailed explanation of how electron deficiency in Ru metals affects K⁺ ion repulsion and water layer structure, which offers a clear and insightful mechanistic understanding for the enhanced NO₃RR and inhibited HER. Totally, this is a highly solid and innovative study on electrocatalytic NO₃RR, which will deepen the understanding of electrocatalytic processes and ions behaviors in the electric double layer, opening up new avenues for catalyst design in electrocatalysis. Therefore, I suggest the acceptance of the manuscript after the authors address the following minor points.

1. What are the key factors that stabilize the 2D Ru metal on the support during electrocatalysis?
2. How to understand the contact potential difference in Figure 2i? Why does a smaller ΔV indicate electrons flow from the Ru metals to the NC support?
3. The BET analysis of various catalysts should be added to exclude the effect of specific surface area.
4. The specific proportions of the three types interfacial water in Figure 5a-b should be provided. Similarly, the energy of each step in Figure 4c should be added.
5. In the potentiostatic tests (Figure S20), the current density increased obviously. Please explain this phenomenon.

Reviewer #3

(Remarks to the Author)

This manuscript by Prof. Li et al. describes an electron deficiency strategy to promote electrochemical NO₃-RR performance. The electron deficiency of Ru metals could boost the repelling effect of K⁺ ions in the electric double layer, which depresses HER and accelerates the penetration of NO₃⁻. The resulted catalyst exhibits excellent NO₃-RR performance with the NH₃ yield rate of 74.8 mg/cm²-h and FE of 94%. However, the current work is NOT significant at current stage and the stability test is not long enough. Further questions should be addressed more clearly.

1. Can this electron deficiency strategy be applied for other metals catalysts such as Pd, Co, and Ni et al.? The authors should provide more evidence to support that Ru is the best choice for electron deficiency effect.
2. In this manuscript the role of K⁺ ions is important. How about other alkali cations such as Li⁺, Na⁺, and Cs⁺? Could they display similar effect in nitrate reduction? The authors should demonstrate this point more clearly.
3. The concentration of KNO₃ solution is 1 M, which is too high for nitrate-to-ammonia conversion. Performance evaluation with lower nitrate concentration should also be done in this work.
4. What is the pH value of the electrolyte after 1 hour test? Could the change of pH affect the NO₃-RR performance and the mechanism of repelling effect of K⁺ ions?
5. H₂ product and other nitrate reduction products such as NO₂⁻, N₂, and N₂H₄ should be tested and quantified.
6. In Line 191, the stability of 6 cycles is too short for metal catalysts. A stability test that is longer than 20 hours is suggested. What's more, characterizations such as XRD, TEM of used samples, and ICP results of the electrolyte after stability test should be provided.
7. In Line 214, literatures for the NO₃-RR performance comparison in Figure 4f-g are few. The authors are suggested to compare their work with other NO₃-RR works using Cu, Co, Pd, Fe et al. as catalysts.
8. In Line 290, stability test at ampere-level is just 6 hours, which is also too short. The authors are suggested to provide the stability test longer than 100 hours.

Version 1:

Reviewer comments:

Reviewer #1

(Remarks to the Author)

The concerns raised by reviewer are well answered. The current manuscript can be published in Nature communications.

Reviewer #2

(Remarks to the Author)

The authors have addressed all of my concerns. The manuscript can be accepted for publication now.

Reviewer #3

(Remarks to the Author)

The authors have addressed my concerns. I recommend to publish the work in Nature Communications.

Responses (R) to Reviewers' Comments (C)

Reviewer #1:

This manuscript focuses on a critical challenge in ammonia synthesis via electrochemical nitrate reduction reaction powered by renewable electricity. The authors reported an important and interesting K^+ ions repelling effect on electron-deficient Ru surface, which is totally different from the previously reported cation effects among different cations. The authors synthesis electron-deficient Ru catalysts with unique 2D Ru metal structure, which have been well characterized and analyzed. In this way, the as-formed 2D-Ru/NC catalyst with pronounced electron deficiency demonstrated excellent NO_3RR performance, especially in additive-free nitrate solution, surpassing that of the reported catalyst in literatures. To prove the crucial role of the repelling effect, detailed in situ spectra and theoretical calculation results gave in-depth mechanistic clarity that the electron-deficient Ru repels the hydrated K^+ ions away from the surface and decreases the density of interfacial water, which would depress hydrogen evolution but promote NO_3RR . Overall, the approach of turning the electron deficiency to modify interfacial water via K^+ ion repelling effect is highly innovative and inspiring, providing valuable insights for future catalyst design in electrocatalysis. In general, I believe this is an excellent work. To facilitate publication, the manuscript should undergo minor revision to address the following points.

C1: The in situ Raman spectra are important evidences for mechanistic clarity. The instrument parameters and experimental procedures about spectra test should be more detailed. Please provide more information.

R1: Thanks for your kind reminding. The *in situ* Raman measurements were carried out by Renishaw inVia Qontor Raman microscope. The excitation wavelength of the semiconductor laser was 525 nm, and the laser power was 2.5 mW. All the Raman measurements were performed with a 50× microscope objective. A H-type three-electrode Raman cell with an embedded quartz window was used for Raman experiments. A carbon cloth loaded with catalyst (loading: 1 mg cm^{-2}) was used as a working electrode. 1 M KNO_3 solution was used as electrolyte. Before Raman test, the working electrode underwent a potentiostatic test to ensure the hydrophilicity. The Vertex.C potentiostat was used to control the potential during the Raman test. We have added these details about Raman spectra test in revised supplementary information.

C2: To prove the stability of synthesized catalyst, the author only provided the Ru 3p XPS spectra of used catalyst. The stability of NC support is also crucial for whole catalyst. Please provide the XPS result of the NC support to confirm its stability.

R2: Thanks for your kind reminding. We have added the C 1s and N 1s XPS results of the NC support in Figure S28 of the revised supplementary information to address reviewer's concern on the structural stability of support.

C3: The reported 2D-Ru/NC catalyst was synthesized by in situ electrochemical reduction method. To highlight the importance of as-formed 2D Ru structure and enhanced integrity and credibility of this work, the NO₃RR performance of an a-Ru/NC electrode should also be tested for comparison.

R3: Thoughtful comments. The NO₃RR current densities of a-Ru/NC were significantly lower than that of 2D-Ru/NC (Figure S18), indicating the vital role of the as-formed 2D Ru sheets in NO₃RR process. We added these data in revised supplementary information.

C4: What are the key factors that contribute to the well-maintained Faradic efficiency values and yields of 2D-Ru/NC catalyst under ampere-level current density? Please provide a more detailed explanation.

R4: Kind suggestions. The enhanced electron exchange between 2D Ru and NC support resulted in stronger interfacial interactions, which prevent the growth of Ru metals in the Z direction and stabilizes the 2D structure of Ru metals to keep from possible aggregation or leaching during electrocatalytic process. To further demonstrate the stability of the catalyst, we conducted a long-term stability test for more than 120 hours (Figure S38) under ampere-level current density in a continuous flow H-type cell. The well-maintained working voltage without obvious Ru leaching (Table S1) again speaks for the good stability of the 2D-Ru/NC electrode even at ampere level. According to the reviewer's kind suggestions, we added detailed discussion in the revised version.

Again, we highly appreciate the reviewer's constructive comments and the great efforts to make our paper better.

Reviewer #2:

In this paper, the authors present an important and attractive breakthrough in electrocatalytic nitrate reduction (NO₃RR) for ammonia synthesis as a viable alternative to traditional Haber-Bosch processes. The reported 2D-Ru/NC catalyst achieves ampere-level NO₃RR reduction in additive-free nitrate solution with record-high yields and well-maintained Faradic efficiency. Both the characterization of the materials and the electrocatalytic performance tests are carried out properly and rigorously. Based on the electronic structures and catalytic performance of catalysts, as well as various characterizations and ab initio molecular dynamics results, the authors proposed K⁺ ion repelling effect to explain the enhanced NO₃RR performance. The key innovation of this work lies in the detailed explanation of how electron deficiency in Ru metals affects K⁺ ion repulsion and water layer structure, which offers a clear and insightful mechanistic understanding for the enhanced NO₃RR and inhibited HER. Totally, this is a highly solid and innovative study on electrocatalytic NO₃RR, which will deepen the understanding of electrocatalytic processes and ions behaviors in the electric double layer, opening up new avenues for catalyst design in electrocatalysis. Therefore, I suggest the acceptance of the manuscript after the authors address the following minor points.

C1: What are the key factors that stabilize the 2D Ru metal on the support during electrocatalysis?

R1: Kind suggestions. The enhanced electron exchange between 2D Ru and NC support resulted in stronger interfacial interactions, which prevent the growth of Ru metals in the Z direction and stabilizes the 2D structure of Ru metals to keep from possible aggregation or leaching during electrocatalytic process. To further demonstrate the stability of the catalyst, we conducted a long-term stability test for more than 120 hours (Figure S38) under ampere-level current density in a continuous flow H-type cell. The well-maintained working voltage without obvious Ru leaching (Table S1) again speaks for the good stability of the 2D-Ru/NC electrode even at ampere level. We added detailed explanation on this point in the revised version to clarify the reviewer's concern.

C2. How to understand the contact potential difference in Figure 2i? Why does a smaller ΔV indicate electrons flow from the Ru metals to the NC support?

R2: Thoughtful comments. The electrons flowing from the Ru metals to the NC support would result in lower surface potential of the NC support. Thus, as compared with bare NC, there was a smaller contact potential difference between 2D-Ru/NC sample and Si substrate. According to the reviewer's kind suggestions, we added more discussion on the contact potential difference in the revised version.

C3. The BET analysis of various catalysts should be added to exclude the effect of specific surface area.

R3: Thanks for your kind reminding. We have added the BET analysis results of the 2D-Ru/NC, np-Ru/NC, and bare NC support in Figure S7 of the revised version. The BET surface areas of 2D-Ru/NC and np-Ru/NC are very close, which could exclude

the significant effect of specific surface area on the final performance of NO₃RR.

C4. The specific proportions of the three types interfacial water in Figure 5a-b should be provided. Similarly, the energy of each step in Figure 4c should be added.

R4: Thanks for your kind reminding. We have added these data as Table S4 and Table S7 in revised supplementary information.

C5. In the potentiostatic tests (Figure S20), the current density increased obviously. Please explain this phenomenon.

R5: Thoughtful comments. The NO₃RR process is always accompanied by the formation of OH⁻ to accelerate the NO₃RR process due to the change in pH value of the electrolyte. Indeed, the pH value of the cathode electrolyte becomes as high as 10.2 within 3 minutes and finally to around 13.0 after two hours (Figure S29) during the potentiostatic test at -1.1 V vs. RHE. More importantly, the unique K⁺ repelling effect cations with all anions including NO₃⁻ and OH⁻ ions enriched on the electron-deficient Ru surface (Figure 5c-d and S34-35) further benefits the NO₃RR process with gradually promoted current output at the earlier stage of the reaction. We also discussed in detail on this point in the main text and Figure caption of Figure S29 to explain the increased current density.

We highly appreciate the reviewer's kind suggestions and thoughtful comments to make our paper complete and better.

Reviewer #3 (Remarks to the Author):

This manuscript by Prof. Li et al. describes an electron deficiency strategy to promote electrochemical NO₃⁻RR performance. The electron deficiency of Ru metals could boost the repelling effect of K⁺ ions in the electric double layer, which depresses HER and accelerates the penetration of NO₃⁻. The resulted catalyst exhibits excellent NO₃⁻RR performance with the NH₃ yield rate of 74.8 mg cm⁻² h⁻¹ and FE of 94%. However, the current work is NOT significant at current stage and the stability test is not long enough. Further questions should be addressed more clearly.

C1. Can this electron deficiency strategy be applied for other metals catalysts such as Pd, Co, and Ni et al.? The authors should provide more evidence to support that Ru is the best choice for electron deficiency effect.

R1: Thoughtful comments. Principally, such an electron deficiency strategy induced by the difference between the work functions of metal and NC support is applicable to different metals. Experimental results in the literature (for example please see *J. Am. Chem. Soc.* 2023, 146, 668-676, *Nat. Commun.* 2024, 15, 6278) and from our group (Figure R1) have shown that Ru metal-based electrocatalyst could provide very high current output for NO₃RR and was thus selected as the model catalyst with electron deficiency effect for possible application in the ampere-level nitrate reduction. We also added more discussion on the superiority of electron deficient Ru in the revised version.

Figure R1. **a** i-t curves of Ru, Pd, Pt, Rh and Ir-based catalysts in 1 M KNO₃ at -1.1 V vs. RHE. **b** NH₃ yields and FE values of Ru, Pd, Pt, Rh and Ir-based catalysts at -1.1 V vs. RHE for NO₃RR.

C2. In this manuscript the role of K⁺ ions is important. How about other alkali cations such as Li⁺, Na⁺, and Cs⁺? Could they display similar effect in nitrate reduction? The authors should demonstrate this point more clearly.

R2: Exactly, Li⁺, Na⁺, K⁺ and Cs⁺ may also display similar cation effect in various electrochemical reactions as described in the literature (For example see *Angew. Chem. Int. Ed.*, 2024, 63, e202408382, *Nat. Catal.* 2024, 7, 807–817, *Nat. Catal.* 2022, 5, 923–933). Even though the AIMD simulation results (Figure S36) indicate that the unique cation repelling effect of electron-deficient Ru surface is similar for Li⁺, Na⁺, and Cs⁺, K⁺ is the best choice among the four cations to give the highest NH₃ yield for NO₃RR under fixed conditions in this work (Figure S24). As a result, we choose K⁺ as

the main focus in this study. We also provided the reason for selecting K^+ ion in the revised version.

C3. *The concentration of KNO_3 solution is 1 M, which is too high for nitrate-to-ammonia conversion. Performance evaluation with lower nitrate concentration should also be done in this work.*

R3: Thoughtful comments. We tested the performance of our 2D-Ru/NC electrode in pure nitrate solutions ranging from 0.1 M to 1 M with Faradaic efficiencies between 98–99% to ammonia (Figure S23a-b of revised supplementary information). When the nitrate concentration is further reduced, the reaction current becomes too low (Figure S23c of revised supplementary information) due to the insufficient electrolyte in the solution. Thus, by adding K_2SO_4 as supporting electrolyte, high Faradaic efficiency (97–99%) can still be achieved at ultra-low nitrate concentrations (0.001–0.01 M), with NH_3 yield from 1.1 to 3.4 $mg\ cm^{-2}\ h^{-1}$ (Figure S23d-e of revised supplementary information). We added experimental results (Figure S23) and detailed explanation on this point in the revised version to clarify the reviewer's concern.

C4. *What is the pH value of the electrolyte after 1 hour test? Could the change of pH affect the NO_3^- RR performance and the mechanism of repelling effect of K^+ ions?*

R4: Expert comments. The NO_3^- RR process is always accompanied by the formation of OH^- . During the potentiostatic test at $-1.1\ V$ vs. RHE, the pH value of the cathode electrolyte becomes as high as 10.2 within 3 minutes and finally to around 13.0 after two hours (Figure S29 of revised supplementary information). Therefore, the *in-situ* spectroscopy tests are inevitably accompanied by an increase in pH, and the results to some extent reflect the influence of alkaline environment. That's why we simulate K^+ repelling effect in the presence of OH^- ion in the AIMD simulation model (Figure 5c). The AIMD simulation results showed a fast departure of hydrated K^+ ions away from the electrode surface, while the OH^- ions are not repelled away from the electron-deficient Ru surface (Figure S33). We also added the measured pH values to the updated Figure S29 of revised supplementary information and described the presence of OH^- ion in all reactions and also in the AIMD simulation model to clarify the reviewer's concern on this point.

C5. *H_2 product and other nitrate reduction products such as NO_2^- , N_2 , and N_2H_4 should be tested and quantified.*

R5: Thanks for your kind reminding. We quantified the amounts of NO_2^- and N_2H_4 using colorimetric methods (Figure S21a-d). And H_2 and N_2 were determined by gas chromatography (GC) (Figure S21e-h). After the potentiostatic test at $-1.1\ V$ vs. RHE, the 2D-Ru/NC catalyst achieved high FE value for NH_3 ($> 99\%$) and negligible FE value for NO_2^- ($\sim 0.2\%$), while no gaseous products and N_2H_4 were detected (Figure S21i). We added detailed detection method for various products and the results in revised supplementary information to clarify the reviewer's concern on this issue.

C6. *In Line 191, the stability of 6 cycles is too short for metal catalysts. A stability test*

that is longer than 20 hours is suggested. What's more, characterizations such as XRD, TEM of used samples, and ICP results of the electrolyte after stability test should be provided.

R6: Thanks for your kind reminding. We reconducted the cycling experiments of the 2D-Ru/NC catalyst for a total reaction time exceeding 20 hours. Again, the yields and Faradaic efficiency values for ammonia could be well maintained after 12 cycles of reuses. We also provided the XRD patterns (Figure S26 of revised supplementary information), TEM images (Figure S27 of revised supplementary information) and XPS analysis results (Figure S28 of revised supplementary information) of fresh and spent 2D-Ru/NC electrodes. Also, Ru leaching is not detectable according to the ICP-ACS analysis result (Table S1). We updated the new cycling data in Figure 3d and structure characterization results to Figure S26-28 of the revised version to demonstrate the durability of the catalytic.

C7. In Line 214, literatures for the NO₃RR performance comparison in Figure 4f-g are few. The authors are suggested to compare their work with other NO₃RR works using Cu, Co, Pd, Fe et al. as catalysts.

R7: Thanks for your kind suggestion. Due to the challenge in NO₃RR in pure nitrate solution with a limited amount of literature at the moment, we have presented all the data we could find for the NO₃RR performance comparison in pure nitrate solution. Also, we added more data for performance comparison using other metals as catalysts in neutral electrolytes. All the data were added in revised Figure 4f and Table S5 to help readers gain a better recognition of our catalysts' performance.

C8. In Line 290, stability test at ampere-level is just 6 hours, which is also too short. The authors are suggested to provide the stability test longer than 100 hours.

R8: Thanks for your kind suggestion. We conducted a long-term stability test for more than 120 hours (Figure S38) under ampere-level current density in a continuous flow H-type cell. The well-maintained working voltage without obvious Ru leaching (Table S1) again speaks for the good stability of the 2D-Ru/NC electrode even at ampere level. We added detailed results (Figure S38 and Table S1) and explanation on this point in the revised version to clarify the reviewer's concern.

Again, we highly appreciate the constructive comments and the great efforts of the reviewer to improve our paper.